# Unruh Entropy of a Schwarzschild Black Hole

Maksym Teslyk [1,2] , Olena Teslyk [2], Larissa Bravina [1] and Evgeny Zabrodin [1,3,*]

1   Department of Physics, University of Oslo, PB 1048 Blindern, N-0316 Oslo, Norway;
    machur@ukr.net (M.T.); larissa.bravina@fys.uio.no (L.B.)
2   Faculty of Physics, Taras Shevchenko National University of Kyiv, UA-01033 Kyiv, Ukraine;
    teslyk.olena@knu.ua
2   Skobeltsyn Institute of Nuclear Physics, Moscow State University, RU-119991 Moscow, Russia
*   Correspondence: zabrodin@fys.uio.no

**Abstract:** The entropy produced by Unruh radiation is estimated and compared to the entropy of a Schwarzschild black hole. We simulate a spherical system of mass $M$ by a set of Unruh horizons and estimate the total entropy of the outgoing radiation. Dependence on the mass and spin of the emitted particles is taken into account. The obtained results can be easily extended to any other intrinsic degrees of freedom of outgoing particles. The ratio of Unruh entropy to the Schwarzschild black hole entropy is derived in exact analytical form. For large black holes, this ratio exhibits high susceptibility to quantum numbers, e.g., spin $s$, of emitted quanta and varies from 0% for $s = 0$ to 19.0% for $s = 5/2$.

**Keywords:** Unruh effect; black hole; information entropy; entanglement entropy

## 1. Introduction

As known from general relativity, any emission from a collapsing body having mass $M$ gradually decreases while the matter falls. The energy required for the particle to escape from the surface diverges reaching the Schwarzschild radius (in Planck units) $r = 2M$, thus any radiation vanishes at the horizon's surface. After that any future light cone becomes directed inwards, and the collapsar transforms into a black hole. Being trapped below the event horizon, one needs a superluminal velocity to break the barrier. In relativity, a black hole is indeed black. Surprisingly, such objects exhibit behavior typical for thermal systems. It turned out that a black hole has entropy proportional to its area [1]. Moreover, associating its surface gravity and area surface to temperature and entropy correspondingly, one arrives at the laws reminiscent of those from thermodynamics, namely [2]:

0.   The surface gravity is the same all over the black hole's event horizon.
1.   The change of energy is expressed via the change of its area plus terms similar to work.
2.   The area of its event horizon cannot decrease.
3.   No finite process can eliminate the surface gravity completely.

Here, the number of the law relates to its counterpart from thermodynamics. This allows concluding that, as soon as the black hole cannot emit any radiation, its temperature should be zero. Therefore, one cannot expect the hole to be in equilibrium, because there is no two-way communication to provide any kind of detailed balance. And this seemed to be an unsolvable mystery: how can the black hole have so much in common with a thermal system, if it cannot emit any radiation?

The solution came from an unexpected direction—quantum physics [3]. Due to quantum fluctuations, particle-antiparticle pairs pop out from the vacuum even close to the horizon. One of them may be lucky enough to escape from a black hole, while its counterpart will fall inside. For any outer observer, the process looks like an outgoing thermal radiation. From the quantum point of view, a black hole is not black at all: it can emit radiation at the expense of its own energy.

Having solved one problem, the answer posed many others. As soon as the radiation from the event horizon is thermal, there is no chance for any information to escape. The emitted particles should be described not as a pure, but rather as a mixed quantum state [4]. This is an essence of the information loss problem that questions the unitary of gravity.

The other issue is the amount of the black hole's entropy $H_{\mathrm{BH}}$. As shown by Hawking, it reads

$$H_{\mathrm{BH}} = A/4\,, \tag{1}$$

where $A$ is the area of event horizon in Planck units. A simple estimate shows that even for the solar mass $H_{\mathrm{BH},\odot} \sim 10^{77}$. Therefore, what processes produce so much entropy? The question is not so simple: for a thermal system, the number of available microstates depends exponentially on its entropy. So, the puzzle becomes a real problem: which degrees of freedom are responsible for the area law, see Equation (1)? To date, many approaches have been proposed to solve the problems.

The obvious suggested solution was to count the relevant microstates. This was performed in different ways. Loop quantum gravity reproduces the area law by the quantization of the black hole's phase space [5,6]. The result strongly depends on the exact value of the Barbero-Immirzi parameter [7–9], but its role still requires further clarification [10].

Alternatively, one can reduce the counting to the statistical problem from polymer physics [11,12], thus assuming entropic origin of gravity [13]. Similarly, the area law can be reproduced within the string theory by estimating the amount of string configurations [14–18].

Quantized black hole spectra offer an elegant solution to the problem of information loss. Namely, particle evaporation should influence the dynamics of its internals under the horizon, thus resulting in less entropic radiation [19] and possible information outflow [20–22].

It looks obvious that it is the event horizon that is responsible for the information loss and the entropy problems. This refers to fruitful and interesting ways of solving the problem such as the brick wall model [23–25], firewalls [26,27] or holography [28–31], thus providing interesting insights into the local thermodynamic properties of the vacuum [32].

The event horizon of a black hole splits a whole space–time into accessible and inaccessible domains. Quantum mechanics states that one should take a partial trace over unobservant degrees of freedom, thus making the entanglement responsible for the entropy production at the horizon [33–38].

Despite that, none of the approaches has yet been widely accepted. Moreover, the necessity of an event horizon to reproduce thermodynamic properties of a black hole can also be disputed [39]. The key problem is the poor knowledge of microscopic description for the space–time with horizons [40,41]. Comprehensive discussions of the issues can be found in reviews [42–46].

In the present study, we analyze the contribution of the Unruh effect [47] to the Bekenstein–Hawking entropy of a Schwarzschild black hole. In order to do this, we simulate the hole by a set of Unruh horizons, and under these we mean the spacetime horizons separating the inside and outside Rindler modes in accelerated frames. A solely geometrical treatment for the entropy problem that origins from both peculiarities of space–time and Hilbert space, is an advantage of the approach.

The paper is organized as follows. Section 2 is devoted to some important issues of probability theory and information theory. Section 3 briefly describes the Unruh radiation mechanism and the properties of its density matrix. Results of the calculation of Unruh entropy that takes into account intrinsic degrees of freedom are presented in Section 4. Section 5 deals with the analysis of asymptotics of the obtained results. Details of the calculations are given in Appendix A. Finally, conclusions are drawn in Section 6.

## 2. Probability and Entropy

Let us consider some discrete non-normalized distribution $\{X\}$, for which $d(x)$ equals to the amount of events with $x$ being observed. The Shannon entropy $H(X)$ for $\{X\}$ may be written as [48]

$$H(X) = -\sum_x \frac{d(x)}{D_X} \ln \frac{d(x)}{D_X} = \ln D_X - \frac{1}{D_X} \sum_x d(x) \ln d(x) , \qquad (2)$$

where $D_X = \sum_x d(x)$. The entropy $H(X)$ quantifies the information one needs to describe $\{X\}$, i.e., the amount of data we are lacking about the system.

For a joint discrete distribution $\{X, Y\}$ with the non-normalized distribution probability $d(x, y)$ the situation looks similar. Its Shannon entropy $H(X, Y)$ reads

$$H(X, Y) = \ln D_{X,Y} - \frac{1}{D_{X,Y}} \sum_{x,y} d(x, y) \ln d(x, y) , \qquad (3)$$

where $D_{X,Y} = \sum_{x,y} d(x, y)$.

At the same time, in the joint case, one may introduce the conditional probability $d(x|y)$ as

$$d(x|y) = \frac{d(x, y)}{d(y)} , \qquad d(y) = \sum_x d(x, y) . \qquad (4)$$

It defines the fraction $x$ from the subset of events with some certain value of $y$. Using Equation (2), the relevant Shannon entropy $H(X|y)$ equals to

$$H(X|y) = \ln D_{X|y} - \frac{1}{D_{X|y}} \sum_x d(x|y). \qquad (5)$$

Finally, substituting Equations (4) and (5) into Equation (3) one obtains

$$H(X, Y) = H(Y) + \langle H(X|y) \rangle_Y = H(X) + \langle H(Y|x) \rangle_X , \qquad (6)$$

where $\langle \alpha \rangle_Y = \sum_y \alpha d(y) / D_Y$.

It can be argued that the information entropy, defined by Equation (2), differs significantly from that in thermodynamics. For any joint distribution describing correlated subsystems, the information entropy quantifies the amount of data encoded in these correlations. It should be proportional to the size of the common boundary between the partitions. Therefore, the information entropy should be governed by some kind of an area law, see, e.g., Ref. [49].

As known from thermodynamics, entropy is an extensive quantity which seems to contradict the conclusion above. To make both quantities compatible, one should take into account conditional distributions. In equilibrium, any correlations vanish, so that the outcomes for $x$ and $y$ become independent. For Equation (6) one obtains then that $H(Y|x) = H(Y)$, and the total entropy reads

$$H(X, Y) = H(X) + H(Y) , \qquad (7)$$

i.e., additivity is restored.

This can be clearly seen in the Boltzmann case. Here momentum distributions $\{X_i\}$, $i = \overline{1, N}$ of $N$ particles are independent, and the total entropy $H_\Sigma(X_1, X_2, \ldots, X_N)$ reads

$$\{X_i\} \equiv \{X\} \quad \Rightarrow \quad H_\Sigma(X_1, X_2, \ldots, X_N) = NH(X) \le NH_B, \qquad (8)$$

thus exhibiting bulk properties. Here, $H_B = \max\{H(X)\}$ stands for the Boltzmann entropy per single particle.

In what follows, we will consider a simplified model that does not contain a direct contribution of conditional distributions. Within the assumptions, Equation (7) suffices,

and the discussion on conditional distributions seems a bit redundant. However, one should keep in mind that any conservation law or interaction leads to correlations between system constituents. So, the detailed discussion above demonstrates a straight recipe for further improvement of the model and reveals the relationships between the information theory approach and thermodynamic equilibrium.

### 3. Unruh Radiation

From here we will use Planck, or natural, units, i.e., $G = c = \hbar = k_B = 1$.

Consider a spherically symmetric system of mass $M$ and some quantum field surrounding it. The field is supposed to be in a pure vacuum state $|0\rangle$ in the free-falling reference frame and to have no influence on the background metric or the frame (quasiclassical approach). This condition implies that the field energy is negligibly small compared to $M$.

Define an accelerated observer moving at acceleration $\vec{a}$, with the norm $|\vec{a}| = a = (4M)^{-1}$. The corresponding non-inertial reference frame is small enough, so that one can neglect any tidal effects.

As was revealed by Unruh [47], the definition of the vacuum depends on the reference frame. In a curved space–time, the emerging horizon splits a whole domain to the inside and outside partitions. Therefore, the state description will be different in each frame, depending on the preferred basis. For $|0\rangle$ it reads [50–52]

$$|0\rangle = \sqrt{\frac{1 - e^{-E/T}}{1 - e^{-NE/T}}} \sum_{n=0}^{N-1} e^{-nE/2T} |n\rangle_{\text{in}} |n\rangle_{\text{out}} \, , \tag{9}$$

where $E$ is the energy of emitted quanta, and $T$ is the Unruh temperature equal to $T = (8\pi M)^{-1} = a(2\pi)^{-1}$. Parameter $N$ determines the number of dimensions for the Hilbert space in a Fock basis. The ket-vectors with subscripts in the *rhs* denote the corresponding Rindler modes with respect to the event horizon.

The Hilbert space dimension $N$ should be understood in the sense that there are $N$ basis vectors corresponding to quanta multiplicity at certain energy $E$, ranging from 0 to $N - 1$. For example, outgoing fermions restrict the value of $N$ by 2. For bosons, one usually sets $N = \infty$ to have a complete Fock basis. However, the correct value should obey the physical laws, including energy conservation. From this point of view, the assumption of infinite multiplicity is excessive, since no real physical system can emit an arbitrary number of particles [53]. So, in what follows we assume that $N$ is finite.

Expression (9) is the Schmidt decomposition, see, e.g., [54], for which the density matrix of outgoing radiation reads

$$\rho_{\text{out}} = \text{Tr}_{\text{in}} |0\rangle \langle 0| = \frac{1 - e^{-E/T}}{1 - e^{-NE/T}} \sum_{n=0}^{N-1} e^{-nE/T} |n\rangle_{\text{out}} \langle n|_{\text{out}} \, . \tag{10}$$

Therefore, a pure vacuum state $|0\rangle$ transforms into a mixed one for the accelerated observer. From this, it is easy to conclude that the only reason for the Unruh effect is geometry. Namely, the horizon arises from the finiteness of the speed of light and the absence of a preferred complete basis in the Hilbert space. The system as a whole is in a pure state and is governed by a unitary evolution. However, the imposed restrictions (horizon) make it impossible to monitor the global space, thus resulting in a mixed state for the radiation.

Each eigenvalue of $\rho_{\text{out}}$ quantifies the probability to detect $n$ outgoing particles at energy $E$ and temperature $T$. From the information theory point of view, one deals with the conditional multiplicity distribution $\{n|N, E/T\}$ at given $N$ and $E/T$. Its von Neumann entropy reads

$$H_U(\rho_{\text{out}}) = H(n|N, E/T) = \sigma(E/T) - \sigma(NE/T) \, , \tag{11}$$

where

$$\sigma(qE/T) = \frac{qE/T}{e^{qE/T} - 1} - \ln\left(1 - e^{-qE/T}\right). \tag{12}$$

Quantity $H(n|N, E/T)$ is an even function of $E/T$. Its asymptotic behavior with respect to $E/T$ is given by

$$\begin{aligned} \lim_{E/T \to 0} H(n|N, E/T) &= \ln N = \max(H), \\ \lim_{E/T \to \infty} H(n|N, E/T) &= 0. \end{aligned} \tag{13}$$

This behavior can be explained as follows. At high temperatures, when the ratio $E/T$ is small, the eigenvalues of $\rho_{\text{out}}$ approach a constant value. Physically it means that one can neglect any correlations induced by the energy conservation. It leads to homogeneous energy distribution of $\{E\}$ at which the entropy reaches its maximum. The other asymptotics applies to particle emission at energies that significantly exceed the source temperature. Being highly unlikely, such processes are exponentially suppressed. Therefore, the asymptotic behavior of Equation (13) is completely determined by the energy conservation.

## 4. Model: Basic Features and Results

The entropy $H_{\text{U}}(\rho_{\text{out}})$ from Equation (11) does not take into account any other degrees of freedom except the multiplicity $n$ and energy $E$ as a parameter. However, the emitted quanta may carry intrinsic degrees of freedom which should influence the phase space of radiation and, consequently, its entropy.

In addition, the contribution of relevant Hilbert subspace might induce additional conservation laws and affect the reference frame. For example, if some emitted particle carries out a non-zeroth spin, the angular momentum conservation dictates the source to change its background metric. Then, the distribution $\{T\}$ over source temperatures should be analyzed also, thus significantly complicating any calculations. Besides, such an extension will contradict the quasiclassical assumption, thus making the whole formalism questionable.

To overcome the problem, we suggest that the intrinsic degrees of freedom have no influence on the background metric, in full accord with the quasiclassical approach. This is valid for a large enough black hole, when imposing any new quantum number $q$ causes negligible correlations. Thus, one may consider the emission probability to be independent of $q$—similar to the analysis below, see Equation (13). So, due to Equation (7), any intrinsic degrees of freedom increase the Unruh entropy as

$$H(Q, \rho_{\text{out}}) = \ln D_Q + H_{\text{U}}(\rho_{\text{out}}), \tag{14}$$

where $D_Q$ is the number of dimensions of Hilbert space describing the relevant degree of freedom.

The Schmidt decomposition Equation (9) is defined for the $D = 1 + 1$ space–time. Any additional spatial dimensions can be omitted with no consequences for the density matrix $\rho_{\text{out}}$. But any Schwarzschild black hole is embedded in a $D = 3 + 1$ space–time. What is the contribution of the lower dimension effect, if any? To answer this, one should take angular degrees of freedom into account.

The Unruh effect has only one certain direction, which is determined by the unit vector $\vec{a}/a$. It can be argued that the Unruh temperature $T$ does not contain such information, since it is completely determined by the acceleration $a$ [47,55,56]:

$$T = \frac{a}{2\pi}, \tag{15}$$

with no vector data inside.

The situation is similar to the black-body radiation. Despite the fact that its emission spectrum carries no information about orientation, there is some other specific direction,

which is determined by the momentum of emitted particles. For a $D = 3 + 1$ source, its radiation consists of similar sources, each generating $H(q, \rho_{\text{out}})$. For the black body, one estimates its total entropy via the corresponding integral in the phase space. For a spherical shape, this makes entropy an extensive quantity, see the discussion at the end of Section 2.

So, outgoing Unruh particles are emitted in some certain direction, which is encoded with $\vec{a}/a$. Assuming that the total Unruh entropy $H_M$ is extensive with respect to the angular degrees of freedom, we obtain

$$H_M(Q, n | N, E/T) = D_{\vec{a}/a} H(Q, \rho_{\text{out}}) \, , \tag{16}$$

where $D_{\vec{a}/a}$ denotes the number of distinguishable directions. Contrary to the phase space of black-body radiation, here we deal with density matrix. This means that $D_{\vec{a}/a}$ is governed by eigenvalues $l$ of angular momentum operator and its projection $l_z$, $-l \leq l_z \leq l$.

As mentioned above, we are considering a spherically symmetric system. Any non-inertial observer will measure the outgoing radiation from the spherically shaped domain having radius $r = 2M$. For this, the angular momentum of the emitted particle is bounded as

$$0 \leq \sqrt{l(l+1)} \leq \sqrt{L(L+1)} = rp = 2M\sqrt{E^2 - m^2} \, , \tag{17}$$

where $m$ is the particle mass and $E$ is its energy. One might argue that $L$ should be an integer, which may not be the case for the *rhs* of Equation (17). To overcome the problem, we assume that

$$2M\sqrt{\frac{E^2 - m^2}{L(L+1)}} = 1 + \varepsilon, \quad \varepsilon \ll 1.$$

Therefore, the number $D_{\vec{a}/a}$ is just the sum over all available $l$ and $l_z$:

$$D_{\vec{a}/a} = \sum_{l=0}^{l=L} \sum_{-l}^{l} = (L+1)^2 = \frac{1}{4}\left(\sqrt{\frac{E^2 - m^2}{4\pi^2 T^2} + 1} + 1\right)^2 . \tag{18}$$

Here we use Equation (17).

Next, one should take the energy distribution $\{E\}$ into account. In the general case, the probability of emitting a particle should be governed by energy conservation. This means that the mass of the black hole shrinks right after the particle leaves the event horizon. Any elementary emission affects the background metric due to the change in the black hole's internal characteristics. Having no quantum gravity to extract the relevant information, we assume the entropy to be additive with respect to energy, similar to Equation (8). This means that the total Unruh entropy can be obtained by integrating the entropy at a certain energy $E$ over an energy distribution $\{E\}$. Also, we propose the interval for the integral to be

$$m \leq E \leq 1 \ll M \, , \tag{19}$$

so that the maximum energy of emitted quanta could not exceed the Planck scale.

Therefore, the total Unruh entropy for the Schwarzschild black hole can be estimated as

$$H_M(Q, n, E | N, T) = \int_m^1 D_{\vec{a}/a} \ln D_Q dE + \int_m^1 D_{\vec{a}/a} H_U(\rho_{\text{out}}) dE \, , \tag{20}$$

where we have used Equation (14), and $D_{\vec{a}/a}$ is determined by Equation (18).

Quantity $H_M(Q, n, E | N, T)$ estimates the entropy produced by the joint distribution $\{Q, n, E | N, T\}$ of Unruh radiation from the Schwarzschild black hole horizon having temperature $T$. Recall that $N$ determines the possible number of particles and is governed by both spin statistics and energy conservation.

Finally, the ratio of the Unruh entropy to the black hole entropy $H_{\text{BH}} = \left(16\pi T^2\right)^{-1}$ reads

$$\frac{H_M(Q, n, E|T)}{H_{\text{BH}}} = Y_Q + Y_{\text{U}} , \tag{21}$$

where $Y_Q$ and $Y_{\text{U}}$ denote the contribution of intrinsic degree of freedom $Q$ and of Unruh effect, respectively. The corresponding quantities are calculated in the Appendix A; analytic expressions for the terms are presented by Equations (A1) and (A5), respectively.

The scaled term $Y_Q$ depends on two parameters: the mass $m$ of emitted particles and the horizon temperature $T$. It is depicted in Figure 1. We depict the plots for small values of $m$, so that to take into account well-established data about spectra of elementary particles. Within the range, one can notice poor entropy susceptibility to $m$, in full accord with the analytical estimates. Note, however, that the dependence on the mass of emitted quanta should be taken into account due to its impact on energy distribution.

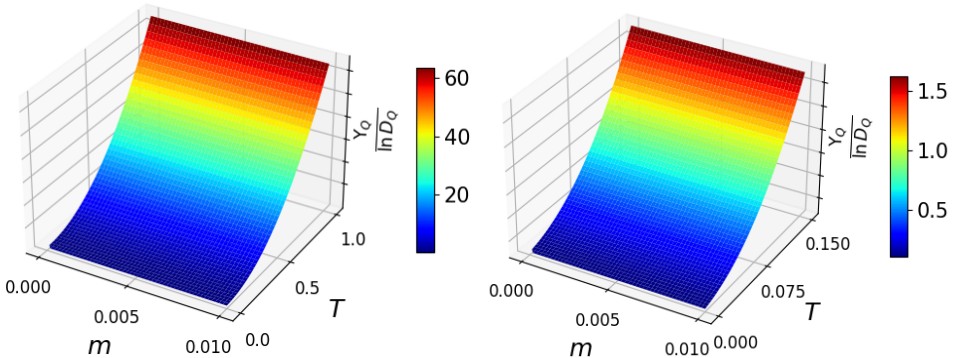

**Figure 1.** (**Left** panel): quantity $Y_Q / \ln D_Q$ from Equation (A1) as a function of $m$ and $T$. (**Right** panel): the same as the left one but for the low-temperature range.

As one can see, the quantity gradually increases with temperature, in full accord with Equation (13). For hot horizons, when $T \gtrsim 0.1$, the ratio $Y_Q / \ln D_Q$ exceeds unity. Recall, that Equation (21) is measured in the units of $H_{\text{BH}}$, so breaking the threshold determines the applicability of the imposed model restrictions.

Contrary to the first term at the *rhs* of Equation (21), the second one turns out to be also a function of $N$. Figure 2 displays the dependence of $Y_{\text{U}}$ on $m, T$ for $N = 2$ (left panel) and on $N, T$ for massless particles (right panel). Again, the contribution of Unruh radiation to the black hole entropy $H_{\text{BH}}$ gradually increases with temperature and $N$. The observed truncation of the plot at $T \lesssim 0.16$ is caused by the numerical precision. This result can be easily deduced by analytic estimates of Equation (A5); see also Equation (13) and the text therein.

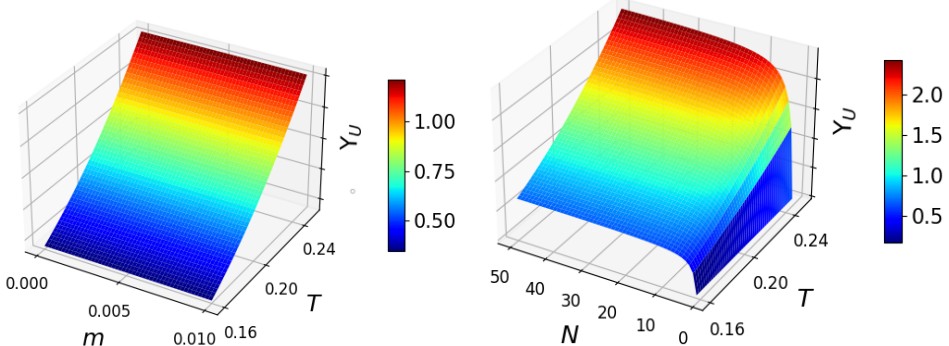

**Figure 2.** (**Left** panel): the term $Y_{\text{U}}$ from Equation (A5) as a function of $m$ and $T$ for $N = 2$. (**Right** panel): the same term as a function of $N$ and $T$ at $m = 0$.

## 5. Asymptotic Analysis

As follows from Equations (A1) and (A5), for hot horizons the ratio given by Equation (21) quickly rises up with increasing $T$. For Planck temperature $T = 1$ it may easily exceed unity. Thus, the proposed model is not valid when $T \approx 1$. It can be argued that this is due to the insensitivity of the emission probability to energy. However, the exact energy distribution $\{E\}$ can be determined from the state hidden below the event horizon, and cannot be treated without gravity quantization.

For the black hole of the stellar scale, the situation is different. In the case $T \to 0$ the Unruh term $Y_U$ vanishes, and $Y_Q$ dominates, as seen from Equations (21), (A1) and (A5):

$$\lim_{T \to 0} \frac{H_M(Q, n, E|N, T)}{H_{BH}} = \frac{1 - 3m^2 + 2m^3}{3\pi} \ln D_Q \geq \frac{\ln D_Q}{3\pi} . \tag{22}$$

The lower bound of Equation (22) for the spin degree of freedom is shown in Figure 3. The total Unruh entropy gradually increases with spin. Here the case $s = 0$ corresponds to the probability of particle emission at the zeroth temperature, when the source cannot emit thermal radiation at all. This result is in line with the previous studies [57,58]. However, even for $s = 1/2$ the ratio reaches 7.35%, and for $s = 5/2$ it touches 19.01%.

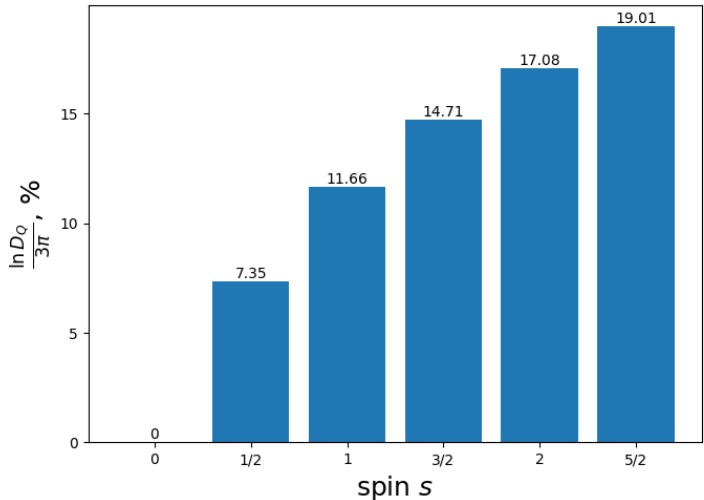

**Figure 3.** Lower bound for $\lim\limits_{T \to 0} \frac{H_M(s,n,E|T)}{H_{BH}}$ at different values of spin.

As follows from Equations (A1) and (A5), massive particles decrease the total Unruh entropy $H_M(Q, n, E|N, T)$. Some terms survive even in the case of small temperatures, see Equation (22). This hints at the influence of correlations in the outgoing radiation, which are encoded by massive particles only. One can consider the effect as some information outflow from the system [19,20] within, e.g., the Page formalism [59,60]. On the other hand, the decrease might originate from the black hole's energy, namely, the more that is deposited in $m$, the less that can be used for entropy production. The effect is insignificant due to the smallness of $m$, since $m \ll 1$ for any elementary particles. Note also that the lower bound in Equation (22) is non-negative for any values of $m$. It should be mentioned that the term surviving in the limit $T \to 0$ is the only one obeying the area law.

## 6. Conclusions

We calculated the total entropy of Unruh radiation from a Schwarzschild black hole of mass $M$. To estimate the contribution of $D = 1 + 1$ effect in $D = 3 + 1$ space–time, the black hole was represented as a set of Unruh horizons. The treatment is valid in the case of the quasiclassical approach when any back-reactions and quantum gravity effects are insignificant. The method allows us to take into account all intrinsic degrees of freedom of

emitted particles, such as spin *s*, charge, etc., together with their mass *m* and multiplicity, defined by the parameter *N*.

As far as information theory is concerned, the Unruh entropy is nothing but the result of inaccessibility to the whole state. This has much in common with the case of limited (partial) access to some chain of bits (qubits), in full accord with the Page formalism. Due to its geometric treatment, the approach implies a unitary evolution for gravity and obeys information conservation.

The calculated entropy $H_M(s, n, E|N, T)$ is represented in exact analytic form. It contains the terms obeying the area law. Surprisingly enough, these terms are governed not by the Unruh effect but by quantum numbers of outgoing particles only.

The entropy $H_M(q, n, E|N, T)$ contains also negative terms, which are proportional to powers of *m* and do not vanish even for $T \to 0$. The role of this effect is still unclear. This can be interpreted as some information outflow from the BH or just a simple consequence of energy deposited by massive particles.

The intrinsic degrees of freedom have a significant impact on the entropy, in full accord with the growth of phase space for the Unruh radiation. The ratio Equation (21) may exceed unity if the relevant Hilbert space is large enough, thus imposing restrictions on the model.

However, this does not mean that the use of the Unruh effect is incorrect for the entropy problem. Recall that the additive entropy behavior is violated by conservation laws. The emerging correlations will induce conditional distributions and, therefore, lower the joint entropy due to Equation (6). For example, spin spectra are not independent because of momentum conservation. Particles with different spins will be correlated, thus resulting in the non-extensive behavior of joint entropy.

We would like to emphasize that the study does not oppose the Unruh entropy to the entropy of the Schwarzschild black hole. We rather expect both these quantities should coincide. However, in order to verify the assumption, one needs to have access to (i) the black hole interior, to extract the energy spectrum distribution $\{E\}$; (ii) total Hilbert space governed by all degrees of freedom of outgoing particles. The last one implies that one knows everything about any correlations among the relevant quantum numbers. All this information should be taken into account; unfortunately, that seems to complicate significantly any analysis.

Note that energy-induced correlations can exactly reproduce the area law, thus favoring evaporation at the event horizon to be unitary [61]. In support of the idea, a recent study [62] shows that quantum hair is crucial for the emission spectrum of a black hole. Therefore, the proper analysis of relevant conditional distributions is of great importance for the black hole entropy problem and deserves further investigation.

**Author Contributions:** Conceptualization, M.T. and E.Z.; methodology, M.T. and O.T.; investigation, M.T. and O.T.; resources, L.B.; data curation, M.T.; writing—original draft preparation, M.T. and O.T.; writing—review and editing, L.B. and E.Z.; visualization, M.T. and O.T.; project administration, L.B.; funding acquisition, L.B. All authors have read and agreed to the published version of the manuscript.

**Funding:** The work was supported by the Norwegian Directorate for Higher Education and Skills (DIKU) under Grant "CPEA-LT-2016/10094—From Strong Interacting Matter to Dark Matter" and by the Norwegian Research Council (NFR) under grant No. 255253/F50—"CERN Heavy Ion Theory".

**Data Availability Statement:** Data sharing is not applicable to this study.

**Acknowledgments:** Fruitful discussions with O. Teryaev and S. Vilchinskii are gratefully acknowledged. Numerical calculations and visualization were made at SAGA (UiO, Oslo) computer cluster facility.

**Conflicts of Interest:** The authors declare no conflict of interests.

### Appendix A. Analytic Expressions for $Y_Q$ and $Y_U$

The term $Y_Q$ can be calculated analytically:

$$
\begin{aligned}
Y_Q &= 4\pi T^2 \int_m^1 \left( \sqrt{\frac{E^2 - m^2}{4\pi^2 T^2} + 1} + 1 \right)^2 dE \ln D_Q \\
&= \left[ \frac{1 - 3m^2 + 2m^3}{3\pi} + 2T\sqrt{1 + 4\pi^2 T^2 - m^2} + 4\pi T^2 (1 - 2m) \right.\\
&\quad \left. + 2T\left(4\pi^2 T^2 - m^2\right) \ln \frac{1 + \sqrt{1 + 4\pi^2 T^2 - m^2}}{2\pi T + m} \right] \ln D_Q .
\end{aligned}
\tag{A1}
$$

The term $Y_U$, see Equations (11) and (18), reads

$$
Y_U = 4\pi T^2 \int_m^1 \left( \sqrt{\frac{E^2 - m^2}{4\pi^2 T^2} + 1} + 1 \right)^2 [\sigma(E/T) - \sigma(NE/T)] dE .
\tag{A2}
$$

It contains incomplete Bose–Einstein integrals and can be calculated as follows. Rewriting Equation (12) as

$$
\sigma(qE/T) = \sum_{k=1}^{\infty} (qE/T + 1/k) e^{-kqE/T}
$$

and using lower incomplete gamma functions $\gamma(\nu, x)$

$$
\gamma(\nu, x) = \int_0^x t^{\nu-1} e^{-t} dt = (\nu - 1)! \left( 1 - e^{-x} \sum_{j=0}^{\nu-1} \frac{x^j}{j!} \right),
$$

for the following integral we obtain

$$
\int_m^1 \sigma(qE/T) E^\nu dE = \frac{T^{\nu+1}}{q^{\nu+1}} \sum_{k=1}^{\infty} \frac{\gamma(\nu+1, x) + \gamma(\nu+2, x)}{k^{\nu+2}} \Bigg|_{x=kqm/T}^{x=kq/T} .
\tag{A3}
$$

Here and below we use the notation

$$
f(x) \Big|_{x=b}^{x=a} = f(a) - f(b) .
\tag{A4}
$$

Next, substituting Equation (A3) into Equation (A2) and using the decomposition

$$
(1 + x)^\alpha = \sum_{n=0}^{\infty} \binom{\alpha}{n} x^n, \quad |x| < 1,
$$

we obtain that

$$
\begin{aligned}
Y_U = \frac{T}{\pi} \Bigg[ &\left( 8\pi^2 T^2 - m^2 \right) \\
&\times \sum_{k=1}^{\infty} \frac{\gamma(1, x) + \gamma(2, x)}{k^2} \left( \Big|_{x=km/T}^{x=k/T} - \frac{1}{N} \Big|_{x=kNm/T}^{x=kN/T} \right) \\
&+ T^2 \sum_{k=1}^{\infty} \frac{\gamma(3, x) + \gamma(4, x)}{k^4} \left( \Big|_{x=km/T}^{x=k/T} - \frac{1}{N^3} \Big|_{x=kNm/T}^{x=kN/T} \right) \\
&+ 8\pi^2 T^2 \sum_{n=0}^{\infty} \binom{1/2}{n} \begin{cases} A_1, & 2\pi T > \sqrt{1 - m^2} \\ A_\mu + B, & 2\pi T < \sqrt{1 - m^2} \end{cases} \Bigg] ,
\end{aligned}
\tag{A5}
$$

where

$$A_\beta = \frac{1}{(2\pi)^{2n}} \sum_{q=0}^{n} \binom{n}{q} (-1)^q \frac{m^{2q}}{T^{2q}} \sum_{k=1}^{\infty} \frac{\gamma(1+2n-2q,x) + \gamma(2+2n-2q,x)}{k^{2+2n-2q}}$$

$$\times \left( \Big|_{x=km/T}^{x=k\beta/T} - \frac{1}{N^{1+2n-2q}} \Big|_{x=kNm/T}^{x=kN\beta/T} \right)$$

$$B = (2\pi)^{2n-1} \sum_{q=0}^{\infty} \binom{\frac{1}{2}-n}{q} (-1)^q \frac{m^{2q}}{T^{2q}} \sum_{k=1}^{\infty} \frac{\gamma(2-2n-2q,x) + \gamma(3-2n-2q,x)}{k^{3-2n-2q}}$$

$$\times \left( \Big|_{x=k\mu/T}^{x=k/T} - \frac{1}{N^{2-2n-2q}} \Big|_{x=kN\mu/T}^{x=kN/T} \right)$$

$$\mu = \sqrt{4\pi^2 T^2 + m^2} \,.$$

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
