# Peer review of "Unruh Entropy of a Schwarzschild Black Hole"

_2571-712X, doi:10.3390/particles6030055_

Round 1

Reviewer 1 Report

The authors calculated the information entropy of a black hole with Unruh radiation, under the no back-reaction assumption and the assumption that the intrinsic degrees of freedom, and Unruh radiation degrees of freedom in 3D space are independent of each other. The topic and the methodology of this paper are suitable for publication in the journal, but there are several points that need further clarification. The language and formulation of the manuscript could be improved to make the story more comprehensive. 

1. I am confused by the concept of Unruh horizon throughout the manuscript, which to my knowledge is not a very common concept. A sentence or two should be added to introduce the concept.

2. P2-L85 Citation for Shannon entropy. 

3. The whole conditional entropy part in Section 2 seems redundant, and equations (3)-(6) seem a bit shaky. If I understood right, only the independent distribution case in equation (7) is used in the discussion of the physics problems in the following sections, and it is well-established enough that it does not need the above justification. If the authors want to keep this part, some concrete citations and more explanatory derivation might help.

4. P4-equation (9) It is confusing when bracket expression is not installed. $n_{\rm in/out}$ are not distinguishable from integers, while they actually represent vectors in the Hilbert space. $0 \rightarrow | 0 \rangle_{\rm kru}$, $n_{\rm in} \rightarrow | n \rangle_{\rm in}$, $n_{\rm out} \rightarrow | n \rangle_{\rm out}$, or any alternative notations that distinguishes them from numbers. The indices denote the different Hilbert spaces those brackets are defined in.

5. From figure 1, 2, it seems that the m-dependence is quite negligible. Any reasoning or comments on this?

1. P1-L12 "Radiation from the surface vanishes completely after reaching some certain radius r = 2M in Planck units" $\rightarrow$ "The energy required for the particle to escape from the surface diverges reaching the Schwarzschild radius $r=2M$, thus vanishes any radiation from the surface.".

2.  P1-L36 "Having solved one problem" $\rightarrow$ "Having one problem solved".

3.  P2-L48 "suggestion" $\rightarrow$ "suggested solution".

4. P4-L130 "Regarding the physical meaning of N, it stands for the largest number of emitted 130 quanta at energy E, which equals N - 1" $\rightarrow$ The Hilbert space dimension $N$ should be understood in the sense that there are $N$ basis vectors corresponding to the number counting of the quanta at certain energy $E$, ranging from $0$ to $N-1$.

5. P6-L210 "we 210 assume the entropy to be additive with respect to energy," + thus the total Unruh entropy can be obtained by integrating the entropy for the degrees of freedom at certain energy over an energy distribution...

Author Response

Please, see the separate PDF file.

Erratum: Reference to Shannon entropy is [48], not [46] 

Reviewer 2 Report

If I am mistaken, I would like to apologise later, but I feel that the authors of this paper must be seriously mistaken about the physics of Hawking radiation.

A particle's energy may become negative via the relativistic rotation of its Killing vector. When a pair of such particles appears next to a black hole's event horizon (Killing horizon), one of them may get drawn in. This rotates its Killing vector so that its energy becomes negative and the pair does not violate energy conservation. This allows them to become real, and this is the most crucial difference to the Unruh effect, which is not defined around the Killing horizon.

Then, the positive particle escapes as Hawking radiation, while the negative-energy particle reduces the black hole's energy. At this moment, there is an obvious entanglement between the inward and outward moving particles.

A researcher have recently carried out local analyses in the vicinity of the black hole horizon(2303.11521). It would be better to consider the mechanism of particle production more carefully.

The author seems to use the term "thermal" in the sense that this entanglement disappears completely at the surface of the black hole, but this is only an assumption. The expression "thermal" also implies that the distribution is determined by the Boltzmann factor. It is not a good idea to be vague about this "assumption" when discussing information and the entropy of black holes, in particular when discussing the material covered in this paper.

The separation of Unruh entropy and black hole entropy is also a very unnatural interpretation, especially when they are discussed at the black hole horizon, by the reason we stated above.

As this difference is the subject of this paper, perhaps a more detailed explanation of the author's views should be given.

For the reasons given above, we consider that the author's claims are ambiguous in the current state. In my personal opinion, the assumptions seem to be quite vague and do not seem to add up in places. This was particularly noticeable in the introduction, and I hope that this will be properly rewritten in the revised version.

 I am not qualified to assess the quality of English in this paper.

Author Response

Please, see the separate PDF file.

Reviewer 3 Report

The Authors investigate the entropy of a Schwarzschild black hole and compared it by the one generated by the Unruh radiation. More in particular, they calculate the total entropy of the Unruh radiation from a Schwarzschild black hole and, with the aim of estimating the contribution of (D = 1 + 1) effect in (D = 3 + 1) space-time dimensions, they depict the black hole was as a collection of Unruh horizons.

The manuscript easily reaches the standards of quality, novelty and robustness for this journal. It is also clearly written and suitably formatted. Almost all the relevant references for this study are fairly quoted.

Before recommending it for publication, the Authors should take care of the following minor points:

Title: style-related suggestion: "Unruh entropy of Schwarzschild black hole" -> "Unruh entropy of a Schwarzschild black hole";

Line 8: please better clarify "laws reminiscent of thermodynamics" means here;

Line 45: style-related suggestion: "what degrees" -> "which degrees";

Line 51: here "Barbero-Immirzi parameter" needs for more specific references.

Author Response

Please see a separate PDF file with our replies.

Round 2

Reviewer 2 Report

I believe that the authors have made their position clear in a necessary and sufficient way with this revision.